# Beau’s Lines and COVID-19; A Systematic Review on Their Association

**DOI:** 10.3390/pathogens13030265

**Published:** 2024-03-20

**Authors:** Aris P. Agouridis, Christina Mastori-Kourmpani, Polyna Antoniou, Paschalis Konstantinou, Evangelos C. Rizos, Constantinos Tsioutis

**Affiliations:** 1School of Medicine, European University Cyprus, 2404 Nicosia, Cyprus; cm191479@students.euc.ac.cy (C.M.-K.); pa192176@students.euc.ac.cy (P.A.); pascalinou@gmail.com (P.K.); vagrizos@gmail.com (E.C.R.); k.tsioutis@euc.ac.cy (C.T.); 2Department of Internal Medicine, German Oncology Center, 4108 Limassol, Cyprus; 3School of Nursing, University of Ioannina, 45500 Ioannina, Greece

**Keywords:** Beau’s lines, COVID-19, SARS-CoV-2, vaccination, nails, skin, long COVID-19

## Abstract

Background: Beau’s lines are transverse grooves in the nail plate that result from transient interruption of the growth of the proximal nail matrix after severe disease. The aim of this study is to systematically report all evidence on the association of Beau’s lines with COVID-19 infection or vaccination against COVID-19. Methods: PubMed and Scopus databases were searched up to January 2024 for articles reporting Beau’s lines associated with COVID-19 infection or vaccination for COVID-19. PROSPERO ID: CRD42024496830. Results: PubMed search identified 299 records while Scopus search identified 18 records. After screening the bibliography, nine studies including 35 cases were included in our systematic review. The studies were reported from different areas around the world. Included studies documented Beau’s lines following COVID-19 vaccination (two studies) or after COVID-19 infection (seven studies). High variability was recorded in onset and resolution times among included cases, averaging 3 months and 6 months after COVID-19 infection, respectively. In the two studies reporting Beau’s lines after vaccination, onset was at 7 days and 6 weeks and resolution occurred after 8 and 17 weeks, respectively. Conclusions: To the best of our knowledge, this is the first systematic review reporting the association of Beau’s lines with COVID-19 infection and vaccination. Severe immune response can result in the formation of these nail disorders. Of importance, Beau’s lines represent a potential indicator of prior severe COVID-19 infection or vaccination for COVID-19, as well as a sign of long COVID-19 syndrome.

## 1. Introduction

Since the first Coronavirus disease (COVID-19) outbreak reported in December 2019, more than 773,119,173 people with COVID-19 have been reported to WHO, affecting people in various ways (https://covid19.who.int/, accessed 5 January 2024) [1]. In brief, infected people can develop a mild to severe illness with or without the need for hospitalization. Common symptoms of the infection are fever, cough, tiredness, and loss of smell or taste, while severe symptoms may include dyspnea, chest pain, confusion, as well as loss of speech or mobility leading to hospitalization or even increased mortality [2,3]. In general, patients with mild to moderate disease exhibit a better prognosis than those with severe to critical infection concerning mortality rates [2,3].

Apart from affecting various systems like respiratory, cardiovascular, and renal, COVID-19 may also affect, less commonly, the skin including nail disorders [4]. These rare cutaneous manifestations may encompass acral areas of erythema-oedema with some vesicles or pustules (pseudo-chilblain), vesicular eruptions, urticarial lesions, maculopapules, livedo, or necrosis [4]. Nail disorders such as transverse leukonychia (Mees’ lines), onycholysis, onychomadesis, onychoschisis, bruising on fingers, onychorrhexis, and Beau’s lines have been described as well [5,6,7].

Beau’s lines (Figure 1 and Figure 2) are transverse grooves in the nail plate that result from transient interruption of the growth of the proximal nail matrix [8]. Beau’s lines have been associated with severe cutaneous inflammatory diseases such as Steven–Johnson syndrome or Kawasaki disease, as well as with malnutrition or exposure to certain medications. In the context of COVID-19, however, little is known [6]. In rare instances, various nail disorders have also been associated with vaccines for COVID-19.

The present study aims to systematically assess the potential association of Beau’s lines with COVID-19 infection or vaccination against COVID-19.

## 2. Materials and Methods

This systematic review was registered on PROSPERO (ID number: CRD42024496830) and adheres to the Preferred Reporting Items for Systematic Reviews and Meta-Analyses (PRISMA) 2020 statement (PRISMA Statement, Ottawa, ON, Canada) [10]. The primary outcome of the present study was to assess the association of Beau’s lines with COVID-19 infection or vaccination against COVID-19.

### 2.1. Search Strategy 

A systematic search was conducted in PubMed and Scopus databases in English until 31 January 2024. To identify the relevant data, we used the following search strategy key: (SARS-CoV-2 or COVID-19 or COVID OR Vaccine OR Vaccination) AND (Beau lines OR Beau’s lines OR Nail*). From the retrieved articles, references were scrutinized to identify additional suitable studies.

### 2.2. Screening and Eligibility

Two authors (C.M-K. and A.P.) were involved in the eligibility screening process. After record deduplication, they excluded non-eligible studies by screening titles and abstracts and evaluated the remaining full texts for eligibility. Any disagreements were resolved upon discussion and joint examination of the proposed articles with a third author (A.P.A.). Studies conducted in humans and written in English were eligible for the systematic analysis. Regarding study design, all study types (RCTs, Case Control, Cohorts, Case Series and Case Reports) were eligible, as well. Studies not meeting the eligibility criteria were excluded. 

### 2.3. Data Extraction

Two authors (C.M.-K. and A.P.) separately extracted data from eligible studies. Data were subsequently added into an electronic document (an Excel spreadsheet) to avoid possible errors in data entry. Reviewer discrepancies were resolved by a third author (A.P.A.), and consensus was reached. Extracted data were reported for the following variables: first author; country; study type; number of patients; population characteristics; severity of COVID-19 infection; comorbidities; sex; age; time of onset of Beau’s lines after infection or vaccination; time to resolution.

### 2.4. Methodological Assessment of Included Studies

The assessment of case reports or case series was performed via the Joanna Briggs Institute (JBI) critical appraisal checklist [11]. Based on the JBI critical appraisal checklist, the items that were evaluated for each study were the patient’s demographic characteristics, patient’s history, patient’s current clinical condition, diagnostic tests or assessment methods and the results, the intervention(s) or treatment procedure(s), post-intervention clinical condition, adverse events (harms) or unanticipated events, and takeaway lessons. The case reports’ assessment was indicated either with a “yes”, which was suggestive of a low risk of bias, or a “no”, which, in any of the included questions, was suggestive of poor overall quality of that case report. In the case, if an item was indicated as an “unclear”, it was suggestive of an unclear or an unknown risk of bias. C.M-K. and A.P. were the 2 independent reviewers that performed a risk-of-bias assessment with any disagreements resolved by consensus with a third reviewer (A.P.A.).

The assessment of the case–control study was performed via the Newcastle–Ottawa scale (NOS) [12]. The 3 main criteria that are taken into consideration included selection, comparability, and exposure. Selection has 4 subcategories, while comparability has 3 subcategories used in the assessment. Due to the lack of an official universal categorization, we searched the literature and concluded that each study could receive a maximum score of 9 points [13]. This was performed based on the literature search where some acceptable options were a score of 7–9, suggesting high-quality studies, a score of 4–6 points, suggesting a high risk for bias, and a score of 0–3, indicating a very high risk for bias [13].

## 3. Results

### 3.1. Study Selection 

In Figure 3, PRISMA flow chart highlights the selection procedure. PubMed search identified 299 records while Scopus search identified 18 records, as well. After duplicate removal and exclusion of irrelevant articles, we examined 12 articles in detail. Finally, nine studies including 35 cases were included in our systematic review.

### 3.2. Study Characteristics

All included studies were published between 2020 and 2024 [9,14,15,16,17,18,19,20,21]. The studies were conducted in different areas around the world, including the USA (three studies), Germany (one study), Canada (one study), Turkey (one study), Brazil (one study), Japan (one study), and Greece (one study). Eight studies were case reports, while one was a case–control study. Seven studies described isolated case reports, one study described two cases, and the case–control study reported twenty-six cases. 

### 3.3. Patient Characteristics

All included studies documented Beau’s lines after COVID-19 vaccination (two studies) [19,21] or after COVID-19 infection (seven studies) [9,14,15,16,17,18,20]. One study reported two children, 2 and 5 years of age, while in the remaining seven isolated cases (four males, three females), the mean age was 60 years old. In the case–control study, no sex or mean age of the twenty-six patients was reported. 

### 3.4. Patient Outcomes

Mean onset time after COVID-19 infection was approximately 3 months, while mean resolution time calculated from the available data was 6 months. Onset in the two studies reporting Beau’s lines after vaccination was 7 days and 6 weeks, respectively, with resolution reported after 2 months and 17 weeks, respectively [19,21]. In the case–control study [9], as well as in two reported cases, the appearance of Beau’s lines was associated with a prior severe COVID-19 infection [14,20]. Table 1 lists the characteristics of included patients. 

### 3.5. Quality Appraisal

The overall quality of the cases was good, as most articles were determined to have a low risk of bias according to the JBI checklist. Table 2 and Table 3 describe and explain the studies’ quality appraisal. In brief, six reports were characterized as low-risk, one report as moderate, and only one as having a high risk of bias. Regarding the case–control report, it was evaluated as a high-quality study.

## 4. Discussion

The current systematic review focuses on the association of Beau’s lines with COVID-19 infection or vaccination. To the best of our knowledge, this is the first systematic approach reporting the potential link between Beau’s lines and COVID-19 infection or vaccination for COVID-19.

Beau’s lines are transverse grooves which extend from one lateral nail fold to the other throughout the nail plate [8]. It is hypothesized that temporary cessation of nail matrix formation or decreased nail plate deposition results in the formation of Beau’s lines adjacent to the proximal nail fold [22]. As mentioned above, Beau’s line formation has been associated with severe inflammatory diseases. Other triggers include malnutrition, chemotherapeutic agents, high fever, and major surgical procedures [8]. They are more commonly observed in adults; nevertheless, they can also appear in infants 4–10 weeks of age due to the stress of delivery as well as in premature infants due to intrauterine stress [23]. Beau’s lines are most prominent on great toenails and thumbs owing to their slower rate of growth.

Regarding pathophysiological mechanisms linking Beau’s lines and COVID-19, little is known. Periods of systemic stress, such as the impact of severe COVID-19 on the body, could trigger their formation. The exact mechanisms linking the Coronavirus infection to these specific nail abnormalities are not fully understood, but it is hypothesized that the immune response and vascular changes associated with COVID-19 may contribute to nail matrix disruptions. Specifically, the virus-induced immune response can trigger inflammation and compromise blood vessel integrity, affecting the nutrient supply to the nail matrix. Consequently, this may lead to nail abnormalities and matrix disruptions, thus manifested as the transverse nail grooves that characterize Beau’s lines.

As is evident from the majority of the included studies [9,15,16,17,18,20], high fever may play a role in triggering the formation of Beau’s lines. A possible explanation on the formation of Beau’s lines during stressful conditions, such as severe COVID-19 with fever, is that during high fever episodes, the body prioritizes vital functions over non-essential ones including nail growth.

In most of the cases included in our systematic review, Beau’s lines were observed 3–4 months after infection, in both males and females as well as in children. Apart from COVID-19 infection, most of the patients were reported to have other comorbidities as well, including thyroid disease (hypothyroidism, hyperthyroidism), iron deficiency anemia, diabetes mellitus, hypertension, hyperlipidemia, allergic rhinitis, rheumatoid arthritis, and osteoarthritis. Therefore, the combination of a systemic illness with an acute infection such as COVID-19 can interfere with normal cell division leading to cessation of normal nail growth and eventually to formation of nail abnormalities such as Beau’s lines.

Similarly, several dermatological conditions have been described in patients with COVID-19 infection, apart from Beau’s lines [4]. Possible pathophysiological mechanisms include the expression of ACE2 receptors by keratinocytes, making the skin sensitive to potential infection, microthrombi formation, cytokine release, and vasculitis. The most frequently reported dermatological conditions associated with COVID-19 are erythematous (multiform) rash, pernio-like acral lesions, urticaria, and varicella-like eruption [4]. In addition, flare-ups of psoriasis and eczema have been observed with COVID-19 infection due to the release of pro-inflammatory cytokines. 

Various other nail findings have also been associated with COVID-19 [5,6,7]. In addition to Beau’s lines, transverse leukonychia, onychomadesis, as well as paronychia associated with chilblain-like lesions, may be encountered in patients with COVID-19. Although chilblain-like lesions represent microcirculatory morphological changes associated with COVID-19, a relationship between Beau’s lines and chilblain-like lesions could not be established through our systematic analysis. Similar pathophysiological mechanisms, however, link these lesions with acute COVID-19, namely inflammation, endothelial cell dysfunction, and hypercoagulability [24]. Moreover, the red half-moon sign, the transverse orange discoloration, and the diffuse red–white nail bed discoloration are microvascular injuries that have been reported and could be linked with COVID-19 [5,6,7].

As observed from our results, Beau’s lines were reported in only two cases after the administration of the COVID-19 vaccine, specifically of the Pfizer-BioNTech vaccine [19,21]. In both cases, Beau’s lines were observed within the first 6 weeks in females aged 64 and 76 years old. The patients had other comorbidities such as rheumatoid arthritis, asthma, and osteoarthritis. As seen in Table 1, these transverse indentations resolved within a few weeks to months. It should be noted that, although Ricardo et al. reported the appearance of Beau’s lines after COVID-19 vaccination, the described patient also had a previous asymptomatic COVID-19 infection 6 months ago [19]. Despite the report of these two cases, further investigation is warranted to establish a possible association between SARS-CoV-2 vaccination and Beau’s lines, as well as the possible causative pathways.

Of note, in the included multicenter case–control study conducted in Turkey, 165 out of 2171 post-COVID-19 patients reported nail disorders [9]. The authors suggested that the development of Beau’s lines, leukonychia, onycholysis, onychomadesis, onychoschisis, bruising on fingers, and onychorrhexis after COVID-19 may be related to a history of severe COVID-19 [9]. Similarly, two of the included studies reported cases following severe COVID-19 infection that required hospitalization in the intensive care unit [14,20]. 

One issue that should be addressed is the possible link of Beau’s lines with long COVID-19 syndrome, a syndrome that has gained attention due to its long-lasting symptoms that affect quality of life and health outcomes. Long COVID-19 syndrome refers to a set of lingering symptoms that persist long after the initial acute phase of COVID-19 infection [25]. While the virus primarily affects the respiratory system, its impact on various organs and systems can lead to a range of complications. According to our results, Beau’s lines have been observed in individuals who have mainly experienced severe COVID-19 illness. In the context of long COVID-19 syndrome, the body’s prolonged immune response and the inflammatory nature of the infection may contribute to such interruptions in the normal growth cycle of the nails [26]. The lines typically appear months after the underlying health event that triggered them, making them a retrospective indicator, not only of the severity of the illness, but also of its protracted course. The presence of Beau’s lines, therefore, serves as a visible reminder of the systemic impact of COVID-19.

No specific treatment is required for resolution of Beau’s lines [5]. As nail growth resumes, the transverse grooves approach the distal free edge of the nail plate. The average rate of nail growth is 0.1 mm/day [27]; therefore, an estimation of the point in time of nail matrix arrest is feasible by measuring the distance between the proximal nail fold and Beau’s lines. These transverse indentations resolve within a few weeks to months [5,6]. However, it is crucial to identify the causative factor and treat the underlying cause, as their appearance could potentially indicate a serious condition. 

This systematic review has several limitations. In brief, the small number of available studies, eight case reports and one case–control study, reporting overall only 35 patients with Beau’s lines following COVID-19 or vaccination against COVID-19, downgrade any robust conclusions. This small number of studies included in the systematic review may be attributed to publication bias. In addition, we were unable to perform a meta-analysis due to lack of data. 

With the increasing COVID-19 infection rates along with the global vaccination campaign, more cases are expected to be reported and vaccination should not be discouraged. Physicians should be aware that recognition of Beau’s lines could indicate a prior severe COVID-19 infection or vaccination for COVID-19, as well as long COVID-19 syndrome. Notably, the appearance of Beau’s lines could signify a potential higher probability for severe re-infection and hospitalization. As research into long COVID-19 syndrome continues, physical signs such as Beau’s lines provide valuable insights into the intricate relationship between viral infections and the body’s physiological responses, emphasizing the need for comprehensive healthcare strategies to address the wide-ranging consequences of COVID-19 [28,29].

## 5. Conclusions

Beau’s lines have been associated with several cutaneous conditions, systemic inflammatory diseases, and infections such as COVID-19. The findings of our systematic review suggest that Beau’s lines, an otherwise negligible clinical sign, may represent a potential indicator for severe immune response. Early identification and careful interpretation of these nail abnormalities can prove to be useful in recognizing patients with higher risk of long-lasting post-COVID-19 disease and in identifying patients with higher risk of severe COVID-19 re-infection. 

## Figures and Tables

**Figure 1 pathogens-13-00265-f001:**
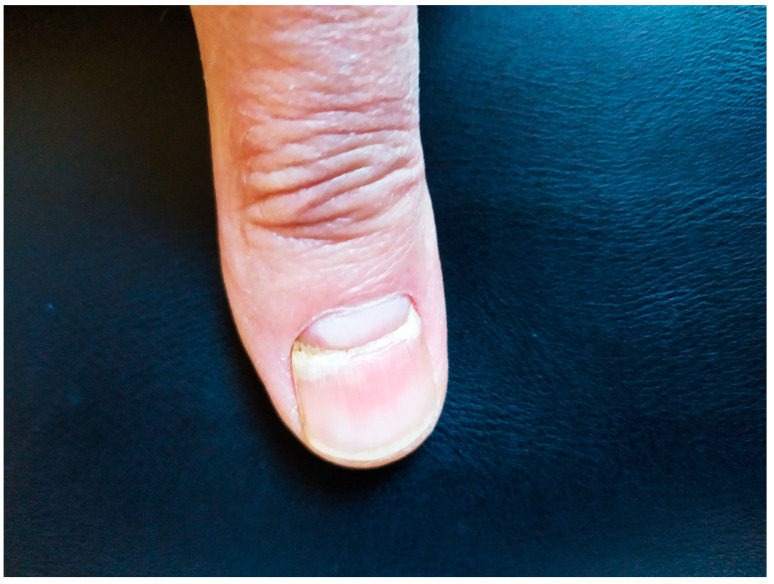
Transverse grooving (Beau’s line) of the second fingernail. Source (reproduced with permission): Agouridis et al. [9], European Journal of Case Reports in Internal Medicine (EJCRIM 2024, https://doi.org/10.12890/2024_004281).

**Figure 2 pathogens-13-00265-f002:**
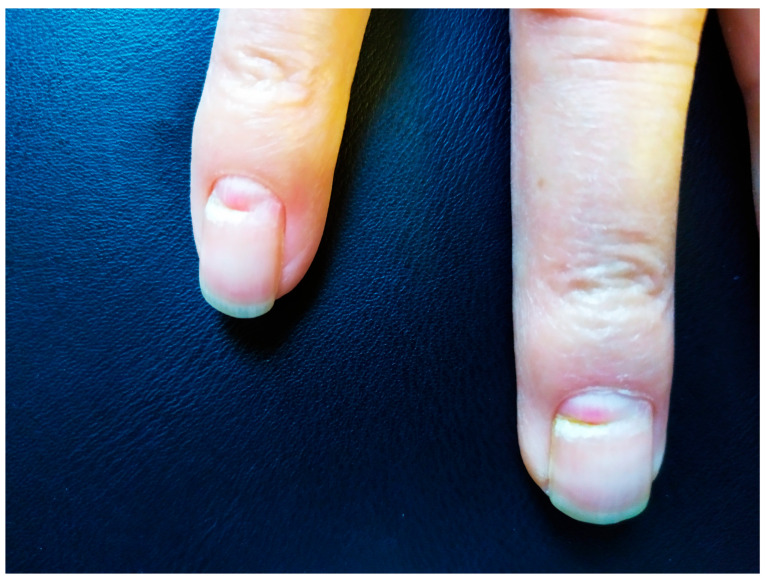
Transverse grooving (Beau’s line) of the fourth and fifth fingernail. Source (reproduced with permission): Agouridis et al. [9], European Journal of Case Reports in Internal Medicine (EJCRIM 2024, https://doi.org/10.12890/2024_004281).

**Figure 3 pathogens-13-00265-f003:**
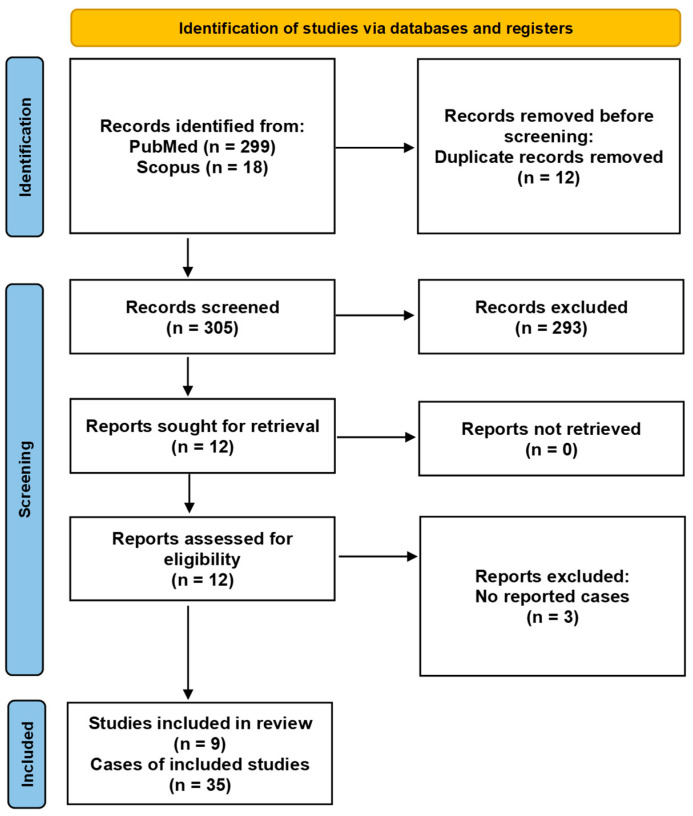
PRISMA flow diagram of articles relating Beau’s lines with COVID-19.

**Table 1 pathogens-13-00265-t001:** Study Characteristics.

Study	Year	Design	Country	Triggering Factor	Cases (n)	Age (Years)	Sex	Comorbidities	Onset After Vaccination	Onset After Infection	Time to Resolution
Agouridis [9]	2024	CR	Greece	Infection	1	65	M	Unremarkable		3–4 months	NR
Kutlu [21]	2023	CCS	Turkey	Infection	26	NR	NR	Thyroid diseases Iron deficiency anemia Hypertension/CVD Diabetes mellitus		1.5–3.5 months	NR
Lam [20]	2022	CR	USA	Vaccination	1	64	F	Rheumatoid Arthritis	6 weeks		17 weeks
Paula [19]	2022	CR	Brazil	Infection	1	58	M	Obesity Hypertension		4 months	10 months
Ricardo [18]	2021	CR	USA	Vaccination	1	76	F	Osteoarthritis Asthma	7 days	6 months	2 months
Deng [17]	2021	CR	USA	Infection	1	41	F	Allergic Rhinitis Diabetes Mellitus Hyperlipidemia		4 months	6–12 months
Wolf [16]	2021	CR	Germany	Infection	2	2 and 5	NR (Children)	NR		3 weeks	4 months
Ide [15]	2020	CR	Japan	Infection	1	68	M	NR		1 month	NR
Alobaida [13]	2020	CR	Canada	Infection	1	45	M	NR		3.5 months	NR

CCS: case–control study; CR: case report; F: female; M: male; NR: information not reported.

**Table 2 pathogens-13-00265-t002:** Reported cases and their risk of bias according to the Joanna Briggs Institute (JBI) Critical Appraisal Checklist for Case Reports [11].

Author	Year	Were Patient’s Demographic Characteristics Clearly Described?	Was the Patient’s History Clearly Described and Presented as a Timeline?	Was the Current Clinical Condition of the Patient on Presentation Clearly Described?	Were Diagnostic Tests or Assessment Methods and the Results Clearly Described?	Was the Intervention(s) or Treatment Procedure(s) Clearly Described?	Was the Post-Intervention Clinical Condition Clearly Described?	Were Adverse Events (Harms) or Unanticipated Events Identified and Described?	Does the Case Report Provide Takeaway Lessons?	Risk of Bias
Agouridis [9]	2024	Yes	Yes	Yes	Yes	Yes	Yes	No	Yes	Low
Lam [20]	2022	Yes	No	Yes	Yes	No	Yes	No	Yes	Moderate
Paula [19]	2022	Yes	No	Yes	Yes	Yes	Yes	No	Yes	Low
Ricardo [18]	2021	Yes	Yes	Yes	Yes	Yes	Yes	No	Yes	Low
Deng [17]	2021	Yes	Yes	Yes	Yes	Yes	Yes	No	Yes	Low
Wolf [16]	2021	Yes	Yes	Yes	Yes	No	Yes	No	Yes	Low
		Yes	Yes	Yes	Yes	No	Yes	No	Yes	
Ide [15]	2020	Yes	No	Yes	Yes	Yes	Yes	No	Yes	Low
Alobaida [13]	2020	No	No	Yes	Yes	Yes	No	No	No	High

**Table 3 pathogens-13-00265-t003:** Quality appraisal of the included studies using the Newcastle–Ottawa Scale (NOS) for Case–Control studies [12].

Study	Selection				Comparability	Exposure			Total Score
	Case Definition	Case Representativeness	Controls Selection	Controls Definition		Ascertainment	Method of Ascertainment	Non-Response Rate	
Kutlu [21]	*	*	-	*	**	*	*	*	8

Star System: A study can be awarded a maximum of one star for each numbered item within the Selection and Exposure categories. A maximum of two stars can be given for Comparability. A score of 7–9 suggests a high-quality study.

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
