# Peer review of "Beau’s Lines and COVID-19; A Systematic Review on Their Association"

_pathogens, 2024, doi:10.3390/pathogens13030265_

Round 1
Reviewer 1 Report
Comments and Suggestions for Authors
The article by Agouridis, A.P. et al. about the relation between Beau´s lines and COVID19 seems scientifically correct and well organized, with recent bibliography supporting the results. Authors performed a good revision, in my opinion. However, I have a major issue:
- As this journal may have a very wide audience. I think there should be an image or an illustration depicting Beau´s lines. The paragraph in lines 50-55 would be the best place to fit this figure.
I have found several minor issues, which can be found below:
- Abstract, line 17. I think you can remove the mention to language restrictions from the abstract (and also the one about the restriction to human, this is obvious). The abstract is a very brief summary. Moreover, in the abstract you state no language restrictions, but in materials you mention that only studies written in English were eligible…
- Abstract, lines 19-20. This sentence should be in past tense. This can be applied to several sentences in the text. Actually, I think you can remove the sentence and leave only: “After screening the bibliography, 9 studies including 35 cases were included…”
- Introduction, page 1, line 41. Please, remove the comma, it can´t be after “than”.
- Introduction, page 1, lines 43-44. Too much commas here.
- Introduction, page 2, line 53. I think you should separate the sentences. You can include a stop after “medications”. Then, you can merge the end of this sentence with the following sentence to talk about what is known for COVID19.
- Materials and methods, page 3, lines 102-109. You refer in this paragraph the “literature”, but you give no citations… You should include various citations to support what is written.
- Results, page 4, lines 126-137. This paragraph is complex to read. You should revise it. May be separating it into two different ones may help.
- Results, page 5, line 143. Please, include “according to the JBI checklist” after “bias”. Remove “Tables may have a footer in Table 2.
- Discussion, page 5, lines 159-161. You should include a reference to support this sentence.
- Discussion, page 6, line 196. May be encountered.
- Discussion, page 6, lines 208-213. 165 out of the 2171 post-COVID-19 patients reported nail disorders. In the following sentence, the authors proposed or suggested. In the last sentence of the paragraph: In this line, 2 of the included studies reported…
- Discussion, page 7, line 231. You should include a reference to support the average nail growth rate.
- Conclusions, page 7, lines 242-264. The last paragraph of discussion and the conclusions paragraph appear very similar. I think you should merge both. You can include either a conclusions section or a final summary in discussion.
Comments on the Quality of English Language
The text has a generally fair grammar / style, with some minor faults, sometimes related to the concordance using verbs or the commas. I suggest a professional review in this respect.
Author Response
Reviewer 1
The article by Agouridis, A.P. et al. about the relation between Beau´s lines and COVID19 seems scientifically correct and well organized, with recent bibliography supporting the results. Authors performed a good revision, in my opinion.
We thank the reviewer for revising our manuscript and stating that our article is well organized.
However, I have a major issue:
- As this journal may have a very wide audience. I think there should be an image or an illustration depicting Beau´s lines. The paragraph in lines 50-55 would be the best place to fit this figure.
We appreciate the reviewer’s comment. The authors have now added a figure of Beau’s lines through the text.
I have found several minor issues, which can be found below:
- Abstract, line 17. I think you can remove the mention to language restrictions from the abstract (and also the one about the restriction to human, this is obvious). The abstract is a very brief summary. Moreover, in the abstract you state no language restrictions, but in materials you mention that only studies written in English were eligible…
The authors thank the reviewer for this comment. We have removed the whole sentence “The qualitative synthesis included studies conducted in humans and had no language restrictions.” from the abstract.
- Abstract, lines 19-20. This sentence should be in past tense. This can be applied to several sentences in the text. Actually, I think you can remove the sentence and leave only: “After screening the bibliography, 9 studies including 35 cases were included…”
The authors made the changes suggested by the reviewer.
- Introduction, page 1, line 41. Please, remove the comma, it can´t be after “than”.
Done
- Introduction, page 1, lines 43-44. Too much commas here.
We have removed the unnecessary commas.
- Introduction, page 2, line 53. I think you should separate the sentences. You can include a stop after “medications”. Then, you can merge the end of this sentence with the following sentence to talk about what is known for COVID19.
The authors made the changes suggested by the reviewer.
- Materials and methods, page 3, lines 102-109. You refer in this paragraph the “literature”, but you give no citations… You should include various citations to support what is written.
We thank the reviewer for this comment. We added the following citation:
Lo, C.K.; Mertz, D.; Loeb, M. Newcastle-Ottawa Scale: comparing reviewers' to authors' assessments. BMC Med Res Methodol 2014, 14, 45, doi:10.1186/1471-2288-14-45.
- Results, page 4, lines 126-137. This paragraph is complex to read. You should revise it. May be separating it into two different ones may help.
The authors made the changes suggested by the reviewer.
- Results, page 5, line 143. Please, include “according to the JBI checklist” after “bias”.
Done
- Remove “Tables may have a footer in Table 2.
Done
- Discussion, page 5, lines 159-161. You should include a reference to support this sentence.
We included the following references:
Park, J.; Li, K. Images in clinical medicine. Multiple Beau's lines. N Engl J Med 2010, 362, e63, doi:10.1056/NEJMicm0906698.
Kim, B.R.; Yu, D.A.; Lee, S.R.; Lim, S.S.; Mun, J.H. Beau's Lines and Onychomadesis: A Systematic Review of Characteristics and Aetiology. Acta Derm Venereol 2023, 103, adv18251, doi:10.2340/actadv.v103.18251.
- Discussion, page 6, line 196. May be encountered.
Done
- Discussion, page 6, lines 208-213. 165 out of the 2171 post-COVID-19 patients reported nail disorders.
Done
- In the following sentence, the authors proposed or suggested.
Done
- In the last sentence of the paragraph: In this line, 2 of the included studies reported…
Done
- Discussion, page 7, line 231. You should include a reference to support the average nail growth rate.
We included the following reference:
Zaias, N. Anatomy and Physiology. In The Nail in Health and Disease; Springer Netherlands: Dordrecht, 1980; pp. 1-18.
- Conclusions, page 7, lines 242-264. The last paragraph of discussion and the conclusions paragraph appear very similar. I think you should merge both. You can include either a conclusions section or a final summary in discussion.
We appreciate the reviewer’s comment. However, the separate Discussion and Conclusions sections are suggested by the journal’s template. As a result, in the Conclusion section we summarize the systematic review’s outcomes.
Reviewer 2 Report
Comments and Suggestions for Authors
This article is a literature review of Beau’s lines (BL) appearing after COVID-19 and after vaccination to SARS-CoV-2. The authors collected data from 9 relevant papers, reporting a total of 35 patients. The rather limited number of studies and patients are a limitation of this review, as the authors honestly acknowledge in the ‘discussion’. Accordingly, I found that the paper is too long when considering these small figures and also the fact that BL do not represent a major medical problem. Therefore, the paper can be shortened to render it easier to the reader (eg too many data on methodology and demographics are presented which are not necessary in my view, such as table 2, the geographic origin of the studies (lines 120-124 – these are given in table 1, etc). The text also contains some repetitions and superfluous sentences (eg lines 41-42: the fact that patients with more severe infection have worse outcomes than those with mild disease is obvious and needs no specific mention). The word ‘systematically’ (line 63) is in my view useless/redundant.
Other issues that need consideration/improvement:
- Line 18: “PROSPERO ID: CRD42024496830 “ needs no mention in the abstract (it is mentioned line 60)
- Lines 25-26: it would be better to express both delays either in weeks (8-17) or in months (2-4) for reasons of homogeneity with the other delays
- table 1: It is more reasonable to present the articles by increasing (not decreasing) year of publication. In this table, ‘time to resolution’ is not provided for the first case, although this is a case previously published by the same group of authors – could you provide this data?
- BL develop usually after febrile illnesses (including COVID-19). Could the authors establish a relation with the presence of (high) fever in the patients of the literature?
- lines 171-175: the authors hypothesize that vascular alterations could be at the origin of BL. Vasculopathy is also thought to be responsible for one of the commonest lesions linked to COVID-19, i.e. chilblain-like lesions (CBLL). Could the authors establish a relationship between BL and CBLL? In other words, did patients with BL present also CBLL (presumably before the diagnosis of BL?)
- lines 191-192: although several skin lesions have been reported in association with COVID-19, these appear usually after the infection. This is especially true for BL, which appear several weeks/months after the infection (which is confirmed by appropriate virological studies), so that their recognition occurs rather late in the course of COVID-19. The sentence (lines 191-192) is therefore not justified, at least as far as skin lesions and BL are concerned, and should be deleted.
- lines 200-2011: the fact that only 2 cases of BL have been reported after SARS-CoV-2 vaccination speaks against a causal relation between the 2, considering how many million people have received this vaccine; a fortuitous association seems likely (at least, it cannot be excluded). Can we be sure that these patients did not have previously some (other) disease/infection triggering BL ? This should be commented.
-A clinical photograph showing Beau’s line should be provided.
- line 127: Ref. 16 is cited in the text before ref. 15 – please check and renumber the references appropriately. Ref. 20: the journal’s name should be abbreviated (line 313).
Comments on the Quality of English Language
- The English needs some editing (eg line 17: ‘con-ducted’, line 24: replace ‘nail disorders’ by ‘nails', line 257: ‘post-covid’).
Author Response
Reviewer 2
This article is a literature review of Beau’s lines (BL) appearing after COVID-19 and after vaccination to SARS-CoV-2. The authors collected data from 9 relevant papers, reporting a total of 35 patients. The rather limited number of studies and patients are a limitation of this review, as the authors honestly acknowledge in the ‘discussion’. Accordingly, I found that the paper is too long when considering these small figures and also the fact that BL do not represent a major medical problem. Therefore, the paper can be shortened to render it easier to the reader (eg too many data on methodology and demographics are presented which are not necessary in my view, such as table 2, the geographic origin of the studies (lines 120-124 – these are given in table 1, etc). The text also contains some repetitions and superfluous sentences (eg lines 41-42: the fact that patients with more severe infection have worse outcomes than those with mild disease is obvious and needs no specific mention). The word ‘systematically’ (line 63) is in my view useless/redundant.
We thank the reviewer for revising our manuscript. We have made several changes and added new data through the text, as well as a figure of Beau’s lines. We hope we have covered the reviewer’s suggestions and comments, accordingly.
We totally agree with the reviewer regarding the length of the paper. However, the journal’s policy on reviews regarding the word count is at least 4000 words.
Other issues that need consideration/improvement:
- Line 18: “PROSPERO ID: CRD42024496830“ needs no mention in the abstract (it is mentioned line 60)
We removed the PROSPERO ID from the abstract
- Lines 25-26: it would be better to express both delays either in weeks (8-17) or in months (2-4) for reasons of homogeneity with the other delays
We thank the reviewer for this comment. We have changed the sentence as follows:
In the 2 studies reporting Beau’s lines after vaccination, onset was at 7 days and 6 weeks and resolution occurred after 8 and 17 weeks, respectively.
- table 1: It is more reasonable to present the articles by increasing (not decreasing) year of publication. In this table, ‘time to resolution’ is not provided for the first case, although this is a case previously published by the same group of authors – could you provide this data?
We thank the reviewer for this comment. Indeed, the first case of the table has been previously published by our group. However, we do not report the resolution time in the published article. That is why we do not mention it in the systematic analysis.
- BL develop usually after febrile illnesses (including COVID-19). Could the authors establish a relation with the presence of (high) fever in the patients of the literature?
The authors thank the reviewer for this comment. We have added the following paragraph in the discussion section:
As evident from the majority of the included studies [13-16,18,20], high fever may play a role in triggering the formation of Beau’s lines. A possible explanation on the formation of Beau’s lines during stressful conditions, like fever in COVID-19 patients, is that during high fever episodes the body prioritizes vital functions over non-essential ones like nail growth.
- lines 171-175: the authors hypothesize that vascular alterations could be at the origin of BL. Vasculopathy is also thought to be responsible for one of the commonest lesions linked to COVID-19, i.e. chilblain-like lesions (CBLL). Could the authors establish a relationship between BL and CBLL? In other words, did patients with BL present also CBLL (presumably before the diagnosis of BL?)
The authors could not establish a relationship between Beau’s lines and chilblain-like lesions through their systematic analysis. They have added the following paragraph through the text:
Although, chilblain-like lesions represent microcirculatory morphological changes associated with COVID-19, a relationship between Beau’s lines and chilblain-like lesions could not be established through our systematic analysis. Similar pathophysiological mechanisms, however, link these lesions with acute COVID-19, namely inflammation, endothelial cell dysfunction, and hypercoagulability [24].
Reference:
- Mehta, P.; Bunker, C.B.; Ciurtin, C.; Porter, J.C.; Chambers, R.C.; Papdopoulou, C.; Garthwaite, H.; Hillman, T.; Heightman, M.; Howell, K.J.; et al. Chilblain-like acral lesions in long COVID-19: management and implications for understanding microangiopathy. Lancet Infect Dis 2021, 21, 912, doi:10.1016/s1473-3099(21)00133-x.
- lines 191-192: although several skin lesions have been reported in association with COVID-19, these appear usually after the infection. This is especially true for BL, which appear several weeks/months after the infection (which is confirmed by appropriate virological studies), so that their recognition occurs rather late in the course of COVID-19. The sentence (lines 191-192) is therefore not justified, at least as far as skin lesions and BL are concerned and should be deleted.
The authors made the changes suggested by the reviewer.
- lines 200-2011: the fact that only 2 cases of BL have been reported after SARS-CoV-2 vaccination speaks against a causal relation between the 2, considering how many million people have received this vaccine; a fortuitous association seems likely (at least, it cannot be excluded). Can we be sure that these patients did not have previously some (other) disease/infection triggering BL? This should be commented.
The authors thank the reviewer for this comment. Indeed, the patient described by Ricardo et al. had a previous asymptomatic COVID-19 infection 6 months before the appearance of Beau’s lines. We added the following paragraph:
“It should be noted that, although Ricardo et al. reported the appearance of Beau’s lines after COVID-19 vaccination, the described patient had also a previous asymptomatic COVID-19 infection 6 months ago.”
-A clinical photograph showing Beau’s line should be provided.
We appreciate the reviewer’s comment. The authors have now added a figure of Beau’s lines through the text.
- line 127: Ref. 16 is cited in the text before ref. 15– please check and renumber the references appropriately.
The references have been cited before in the “study characteristics” in line 124.
Ref. 20: the journal’s name should be abbreviated (line 313).
Done
Round 2
Reviewer 1 Report
Comments and Suggestions for Authors
The article by Agouridis, A.P. et al. about the relation between Beau´s lines and COVID19 addressed my suggestions.
I still can find various issues:
- Introduction, Figure 1. Don´t you have an original image? Moreover, the proposed image has quite low resolution. Honestly, this is really deceptive, you should present a really nice image (or images) of Beau lines. This (these) image (s) don´t need to be related with COVID, but should be good and illustrative. May be you can make an illustration…
- Introduction, page 1, lines 43-44. Too much commas here.
- Introduction, page 2, line 53. I think you should separate the sentences. You can include a stop after “medications”. Then, you can merge the end of this sentence with the following sentence to talk about what is known for COVID19.
- Materials and methods, page 3, lines 102-109. You refer in this paragraph the “literature”, but you give no citations… You should include various citations to support what is written.
- Results, page 4. With this new organization, you should remove “outcomes” from title 3.3.
- Results, page 6. You didn´t removed “Tables may have a footer” in Table 2.
- Conclusions, page 7, lines 242-264. It´s acceptable to present a conclusions section, but in that case, you should remove several sentences in the summary at the end of the discussion section. You can merge if you want the final paragraph of discussion with the conclusion, as they are repetitive.
Comments on the Quality of English Language
The text has a generally fair grammar / style, with some minor faults.
